# The frequency, clinical characteristics and outcomes of *Naja* species related injuries in Malaysia consulted to Remote Envenomation Consultancy Services from 2020–2023

**Ahmad Khaldun Ismail**[1]*, **Zhi Xuan Ng**[1], **Syahirah Rezwan Eskandar**[1], **Hamelda Tanisha Ganaprakasam**[1], **Zainalabidin Mohamed Ismail**[2]

**1** Department of Emergency Medicine, Faculty of Medicine, Universiti Kebangsaan Malaysia, Hospital Canselor Tuanku Muhriz, Kuala Lumpur, Malaysia, **2** Emergency and Trauma Department, Hospital Tengku Ampuan Afzan, Kuantan, Pahang, Malaysia

* khaldun_ismail@yahoo.com

## Abstract

*Naja* species bites and envenomation are common in Malaysia. This is a retrospective cohort study of diagnosed *Naja* species cases consulted to Remote Envenomation Consultancy Services (RECS) from 2020 to 2023. This study aimed to identify the frequency, geographical distribution, clinical features, treatments, and outcomes of *Naja* species-related injuries. Data was extracted following the approval of the institutional research ethics committee and all patient's information were kept anonymous. From 4474 RECS consultations, 512 were diagnosed as caused by *Naja* species. *Naja kaouthia* cases were mostly from the northern and central Peninsular Malaysia, while *Naja sumatrana* was recorded nationwide. There were 184 (35.9%) cases diagnosed as undifferentiated *Naja* species bites, 144 (28.1%) *N. sumatrana* bites, 121 (23.6%) *N. kaouthia* bites, 61 (11.9%) venom ophthalmia and 2 (0.4%) involved bites and venom ophthalmia from *N. sumatrana*. The mean age was 36.2 years old (SD ± 20.7), and 69.9% were male. The median bite to door time was 1 h (IQR: 0–2 h). The most frequent anatomical region involved was the lower limb (52.1%). Local envenomation is the commonest manifestation (*n* = 366, 81.2%). Pain (*n* = 386) and swelling (*n* = 310) were frequent signs of local envenomation, while vomiting (*n* = 54) and ptosis (*n* = 37) were commonest signs of systemic envenoming. Antivenom was administered in 157 (30.7%) cases and 78.3% were Thai Red Cross Cobra antivenom. The median time interval for door to receiving the first dose of antivenom was 12 h (IQR: 5.5-14.5 h). Surgical intervention was performed in 53 (11.8%) cases, mostly were for wound debridement. Four deaths were documented and were due to secondary complications. No antivenom usage, morbidity or death following venom ophthalmia incident. These findings highlight the importance of

**Data availability statement:** The authors confirm that all data underlying the findings are fully available without restriction. All relevant data are within the paper and its Supporting Information files.

**Funding:** The author(s) received no specific funding for this work.

**Competing interests:** The authors have declared that no competing interests exist.

expert support for healthcare professionals for early clinical decision-making to reduce complications and enhance outcomes.

## Author summary

Cobra bites in Malaysia are common but rarely studied. The study included over 500 cases of bite injuries caused by *Naja* species across a 4-year period. It reported on the frequency, symptoms, treatment, and outcomes. *Naja sumatrana* and *Naja kaouthia* were the two cobra species indigenous to Malaysia, with different geographical distribution in the country. Most of the diagnosis were classified as undifferentiated *Naja* species based on the clinical features. The median time of bite to arrival at a healthcare facility was 1 h. The lower limb was the most common part of the body involved with pain and swelling, being the most common local symptoms. Local envenomation had better prognosis than systemic envenomation. Antivenom was utilized in 30.7% of cases with the median door to first antivenom was 12 h. Four deaths were reported throughout the study period. This study highlights the importance of adhering to the national guidelines for snakebite envenomation management supported by consultations from experts in the field for optimal care and outcome.

## Introduction

Snakebite envenomation is an important but often under-reported public health hazard in many countries including Malaysia. The *Naja* (cobras) and *Bungarus* (kraits) species are the leading causes of severe envenomation and mortality in Asia [1–3]. Cobras, belonging to the genus *Naja,* has the potential to cause permanent disability and death in humans. Presently, there are 11 cobra species indigenous to South and Southeast Asia, however, the taxonomy of this genus remains unsettled, with ongoing descriptions of new species. In Malaysia, the most common cause for venomous snake bites is from cobra species, *Naja sumatrana* and *Naja kaouthia* (Fig 1) [4–8]. Cobra species possess potent postsynaptic neurotoxic venoms, with some exhibiting cytotoxic and cardiotoxic effects [9–12]. Their venom is rich in three-finger toxins (3FTx), including polypeptides of neurotoxins and cardiotoxins, as well as phospholipase A2 (PLA2) [13,14]. Envenomation can result in post-synaptic blockade and inhibiting neuromuscular transmission, leading to symptoms such as ptosis, ophthalmoplegia, dysphagia and impairment of skeletal muscles contractility and paralysis. [15,16]. Patients may experience nasal voice, difficulty swallowing, and respiratory failure, and generalized flaccid paralysis which can be fatal [17]. In addition, painful progressive tissue swelling, and necrosis (dermonecrosis), are common manifestations [9,18].

Remote Envenomation Consultation Services (RECS) is a specialized online risk management support system that assists doctors in managing clinical toxicology-related cases. It was founded in early 2012 [10]. RECS offers a 24-hour on-call consultation service through a smart text messaging platform while adhering to guidelines set by the Ministry

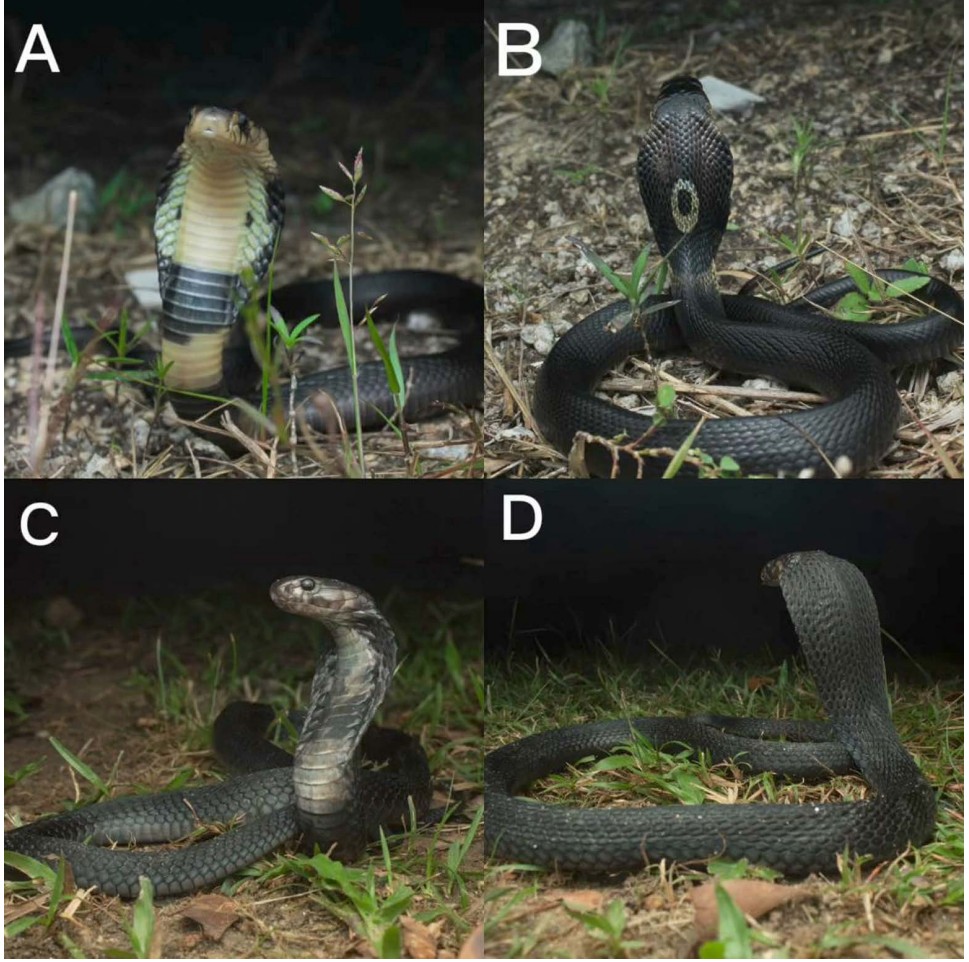

**Fig 1. Morphological features of *N. kaouthia* and *N. sumatrana*.** (A) Ventral view of *N. kaouthia. Image credit: M.K. Arif.* (B) Dorsal view of *N. kaouthia. Image credit: M.K. Arif.* (C) Ventral view of *N. sumatrana. Image credit: M.K. Arif.* (D) Dorsal view of *N. sumatrana. Image credit: M.K. Arif.*

of Health Malaysia (MOH) [19]. RECS consultants are affiliated with the Malaysian Society of Toxinology and the special interest group of the College of Emergency Physician Malaysia. RECS maintains detailed consultation logs that include validated management recommendations and images of the offending animals, with species identification verified by subject matter experts.

Identifying snake species and recognizing the clinical features of envenomation is the main concern among doctors managing snake-related injuries. Poor clinical management may cause mortality and morbidity affecting the cost for health expenditure and human lives. This study is important in facilitating optimal clinical management and at the same time minimizing the cost and wastage of resources. The objective of this study was to analyze the frequency, geographical distribution, clinical features, treatments, and outcomes of *Naja* species-related injuries in Malaysia consulted to RECS.

## Methods

### Ethics statement

Data was collected following the approval of the Universiti Kebangsaan Malaysia research ethics committee (UKM FF-2024–118) and RECS coordinator. This is a retrospective cohort study.

All cases diagnosed as *Naja* species related injuries consulted to and verified by RECS were included. Alleged cobra bite and unidentified snake-related injuries were excluded. A universal sampling method was used to extract data from RECS consultation log and case files. All relevant information from each case files were anonymously documented into a standardized data collection sheet and kept confidential. The diagnosis was confirmed and verified by RECS consultants during each consultation based on the specimen brought to the hospital or the picture of the actual specimen that caused the injury or the clinical presentation and syndromic approach. The diagnosis categories were *Naja kaouthia* bite, *Naja sumatrana* bite, Venom ophthalmia (by *N sumatrana*) and undifferentiated *Naja* sp. bite. Data was analyzed with the assistance of a clinical statistician for the association of length of hospital stay with spectrum of injury and antivenom usage. The comparison between the clinical pattern of injury, management and outcome of both species were calculated using chi-square test. The Mann-Whitney U test was used to analyze the antivenom usage and duration of hospitalization between the two *Naja* species.

## Results

From 4474 RECS consultations, 1070 (23.9%) were in 2020, 1014 (22.7%) in 2021, 1019 (22.8%) in 2022 and 1371 (30.6%) in 2023. The annual frequency of cases appears similar during COVID-19 pandemic phase (2020–2021) and the transition year 2022 (endemic phase). There appears to be an increase of outdoor incidents during the endemic (55.4%) years compared to pandemic (50.4%) years while there were fewer indoor cases during the pandemic (49.6%) years compared to endemic (44.6%) years. A total of 512 *Naja* species related cases were diagnosed within the study period with 254 (49.6%) from 2020-2021 and 258 (50.4%) from 2022-2023. In 2023, *N. sumatrana* related cases appear to be higher than previous years ([Fig 2]).

Most patients were male ($n = 358$, 69.9%). The mean age was 36.2 years (SD ± 20.7) with the youngest was nine months and the oldest was 87 years ([Table 1]). Malaysians ($n = 424$, 82.8%) were the majority. Most injuries were non-occupational related ($n = 433$, 84.6%). The incident mostly occurred in the afternoon ($n = 220$, 43.0%), and in outdoor setting ($n = 265$, 51.8%).

The diagnosis was categorized as: *Naja sumatrana* bite (n = 144, 28.1%); *Naja kaouthia* bite ($n = 121$, 23.6%); Undifferentiated *Naja* species bite ($n = 184$, 36.0%); *Naja sumatrana* venom ophthalmia ($n = 61$, 11.9%); Naja sumatrana bite & venom ophthalmia ($n = 2$, 0.4%). Most of the patients were bitten once ($n = 437$, 96.9%) ([Table 2]). The lower limb was the most common anatomical region affected ($n = 267$, 52.1%). Tourniquet application was the most frequent first aid (pre-hospital) intervention ($n = 154$). The median bite to door (BTD) time was 1 h (IQR: 0–2 h) with the maximum delay of 21 hr. Most *Naja* species bite patients only developed local envenomation ($n = 366$, 81.2%) with 75 (16.6%) of cases developed both local and systemic envenomation. Moderate local pain is the most common sign at presentation ($n = 187$, 36.5%). Eye pain ($n = 56$) and conjunctival injection (red eye) ($n = 36$) were frequent complaints following venom ophthalmia. Bite cases frequently presented with local pain ($n = 386$) and swelling ($n = 310$). The common systemic manifestations following *Naja* species bite were vomiting ($n = 54$) and ptosis ($n = 37$).

Antivenoms were administered to 157 (30.7%) bite cases, with the majority were for undifferentiated *Naja* species cases ($n = 96$, 61.1%), followed by *N. sumatrana* ($n = 35$, 22.3%) and *N. kaouthia* ($n = 26$, 16.6%) ([Table 3]). The commonest antivenom used was Thai Red Cross *Naja kaouthia* monospecific antivenom (NKAV) ($n = 123$, 78.3%) with 75 (61.0%) patients received one dose (5 vials). The median time interval from door to receiving the first dose of antivenom (DAV) was 12 h (IQR: 5.5-14.5 h) with the maximum time interval of 23 h. Intramuscular anti-tetanus toxoid (ATT) was administered in 262 (51.2%) cases. Antibiotics was administered in 157 (30.7%) cases with Amoxicillin/ clavulanate ($n = 76$) being the commonest. Culture and sensitivity were performed in 63 (14.0%) bite cases and the most common organism was *Morganella morganii* ($n = 6$). Complications during hospital stay were documented in 38 (8.2%) patients and mostly were due to wound infection ($n = 34$). Surgical intervention was conducted in 53 (11.8%) bite cases with 48 (90.6%) underwent wound debridement. Among those with systemic envenomation, the most

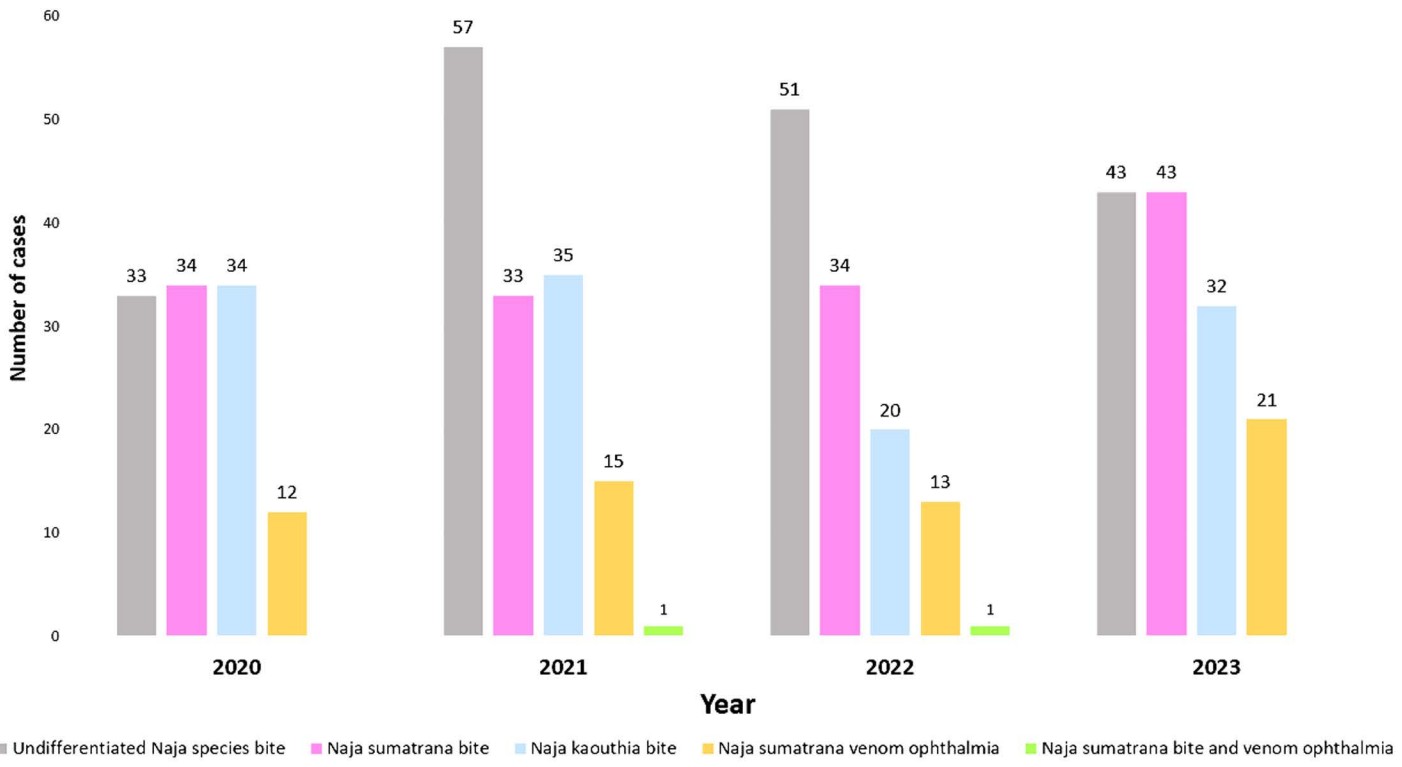

**Fig 2. Frequency and diagnosis of *Naja spp.* related injuries consulted to RECS from 2020-2023.** The consultations with RECS remained similar during COVID-19 pandemic and endemic phase.

frequent emergency resuscitation was for airway intubation and ventilation ($n = 21$). Two patients had cardiopulmonary resuscitation (CPR). Majority of the admission to critical care units were due to systemic envenomation ($n = 8$, 57.1%). Most patients were hospitalized for one day ($n = 153$, 29.9%) and the longest duration of hospitalization was 21 days.

The median length of the stay was 1 (IQR: 1–3). There is an association between the length of hospital stay with the type of envenomation and antivenom usage (S1 Table). A comparative analysis for antivenom usage and length of hospital stay (LOS) was conducted between *N. kaouthia* and *N. sumatrana* envenomation cases using the Mann-Whitney U test (S2 Table). The median total antivenom usage was five vials for both species. However, the interquartile range (IQR) was broader in *N. kaouthia* cases (5.00) compared to *N. sumatrana* (0.00), indicating greater variability in antivenom administration. Despite this, there was no statistically significant difference in the total antivenom usage between the two species (U = 406.50, p = 0.393). The LOS also showed a comparable median duration of 1 day for both species. However, *N. kaouthia* cases had a slightly wider IQR (1.75 days) compared to *N. sumatrana* (1.00 days). The Mann-Whitney U test did not reveal a statistically significant difference in LOS duration between the two groups (U = 5640.00, p = 0.615). Four deaths were documented with three cases were envenomation from undifferentiated *Naja* species and one *N. kaouthia*. Causes of death were due to septic shock, basilar artery occlusion, hypoxic ischemic encephalopathy, and respiratory failure (S3 Table). No antivenom usage, complications or death were recorded for venom ophthalmia cases. There was no significant difference in clinical pattern of injury, management, and outcome between *N. kaouthia* and *N. sumatrana* envenomation ($p > 0.05$) (S4 Table).

**Table 1. Demographic characteristics of *Naja* species related injuries consulted to RECS from 2020–2023.**

| Demographic characteristics | | n (%) |
|---|---|---|
| Gender | Male | 358 (69.9) |
| | Female | 154 (30.1) |
| Age | < 1 | 2 (0.4) |
| (years) | 1-4 | 20 (3.9) |
| | 5-14 | 65 (12.7) |
| | 15-24 | 74 (14.5) |
| | 25-49 | 210 (41.0) |
| | 50-64 | 77 (15.0) |
| | > 65 | 59 (11.5) |
| | Undocumented | 5 (1.0) |
| Nationality | Malaysian | 424 (82.8) |
| | Non-Malaysian | 69 (13.5) |
| | Undocumented | 19 (3.7) |
| Occupation related injury | No | 433 (84.6) |
| | Yes | 62 (12.1) |
| | Undocumented | 17 (3.3) |
| Time (H) | 00:00-06:59 | 35 (6.8) |
| | 07:00-11:59 | 127 (24.8) |
| | 12:00-18:59 | 220 (43.0) |
| | 1900-23:59 | 130 (25.4) |
| Location of incident | Outdoor | 265 (51.8) |
| | Pandemic phase | 122 (50.4) |
| | Endemic phase | 143 (55.4) |
| | Indoor | 235 (45.9) |
| | Pandemic phase | 120 (49.6) |
| | Endemic phase | 115 (44.6) |
| | Undocumented | 12 (2.3) |

*N. kaouthia* cases were mostly consulted from hospitals in the northern and central regions of Peninsular Malaysia while *N. sumatrana* from hospitals throughout the country (Fig 3).

## Discussion

This is the first retrospective cross-sectional study describing the demographic and clinical features of bites by *Naja* species in Malaysia. Previous single-centered studies in Malaysia, though limited, have also highlighted cobras as the predominant cause of venomous snakebites [20–22]. However, none specifically analyze and compare the two indigenous *Naja* species. The findings of this study are comparable with previous studies in Southeast Asia [17,23–25]. In this study, the frequency of *N. sumatrana* injuries was found to be higher compared to *N. kaouthia*. This may reflect geographical differences in *Naja* species distribution between Malaysia and Thailand. The high percentage of undifferentiated *Naja* species cases highlights a common challenge in snakebite management. In many instances of unidentified snakebite, the patient killed and threw away the culprit snake without taking any photographic evidence. This highlights the need to improve snake identification methods in pre-clinical settings, such as increasing the awareness among the public to take pictures of the culprit snake before going to the hospital. The use of rapid immunodiagnostic tests may not be relevant for differentiating the two *Naja* species as the management is the same for both. However, the clinical features of local envenomation in *Naja* species and syndromic approach could distinguish envenoming manifestations from the other elapid bite

**Table 2.** Diagnosis, prehospital care intervention and clinical pattern of *Naja* species related injuries consulted to RECS from 2020–2023.

| Factors | Category | *Naja sumatrana* n (%) | *Naja kaouthia* n (%) | Undifferentiated *Naja* sp. n (%) |
|---|---|---|---|---|
| Number of bites (N = 451) | One | 143 (32.7) | 117 (26.8) | 177 (40.5) |
| | Two | 3 (21.4) | 4 (28.6) | 7 (50) |
| Anatomical region | Lower limb | 81 (30.3) | 63 (23.6) | 123 (46.1) |
| | Upper limb | 65 (35.9) | 57 (31.5) | 59 (32,6) |
| | Head and neck | 1 (50.0) | 1 (50.0) | 0 (0) |
| | Eye(s) | 63 | 0 | 0 |
| First aid intervention | No | 83 (35.8) | 64 (27.6) | 85 (36.6) |
| | Yes | 124 (44.3) | 57 (20.4) | 99 (35.4) |
| Types of envenomation (N = 451) | Local | 130 (35.5) | 107 (29.2) | 129 (35.3) |
| | Systemic | 10 (13.3) | 11 (14.7) | 54 (72.0) |
| | None | 6 (60.0) | 3 (30.0) | 1 (10.0) |
| Pain score at presentation | No pain (0) | 33 (50) | 19 (28.8) | 14 (21.2) |
| | Mild (1–3) | 59 (45.7) | 40 (31.0) | 30 (23.3) |
| | Moderate (4–6) | 73 (39.0) | 37 (19.8) | 77 (41.2) |
| | Severe (7–10) | 30 (29.7) | 19 (18.8) | 52 (51.5) |
| | Unable to assess | 1 (10.0) | 0 (0) | 9 (90.0) |
| | Undocumented | 11 (57.9) | 6 (31.6) | 2 (10.5) |

in Malaysia. Detailed history, examination and serial documentation of progress including the blood tests could further provide clues of the culprit snake.

*Naja kaouthia* is found mainly in the northern, central and the east coast region of Peninsular Malaysia while *N. sumatrana* were found throughout Peninsular Malaysia and East Malaysia [5,6,8]. This distribution aligns with the known geographical range of these species. According to the WHO's Guidelines for the Management of Snakebites (2016), *N. kaouthia* is widely distributed across Southeast Asia, while *N. sumatrana* is more localized to peninsular Malaysia, southern Thailand, and parts of Indonesia [15]. Our findings suggested that *N. kaouthia* is also present in the states of Sarawak, Pahang, Negeri Sembilan and Johor. This finding is unusual and may suggest the possibility of the introduction of non-native species in the localities either by importation or irresponsible release (escapees). The finding of *N. kaouthia* in these states may not affect the clinical management.

A crucial finding is that most were non-occupational. This challenges the traditional stereotype of snakebite as a hazard confined to agricultural workers. The data suggests a more pervasive pattern of peri-domestic and recreational encounters, where human habitats (homes, gardens) and snake habitats increasingly overlap. This has direct implications for public health messaging, which must target the public with information about risks in residential areas, not just farmers in fields. Despite the inappropriate first aid practices, the public appears to understand the urgency of seeking professional medical care.

The data shows that both *N. kaouthia* and *N. sumatrana* primarily caused local envenomation, with a smaller proportion of cases involving systemic effects. This is consistent with the general envenomation profile of Asian cobras [24,26]. Venom ophthalmia appears to be associated only with *N. sumatrana*. This aligns with reports that described *N. sumatrana* as having a unique ability to "spit" venom, leading to ocular injuries [3]. However, no permanent loss of vision was reported in this study. While there was no significant difference in initial pain scores and systemic envenomation from a bite between the two species, *N. kaouthia* bites appeared to require more frequent wound debridement. This may suggest a higher severity of local tissue necrosis from *N. kaouthia* bite. This discrepancy may reflect differences in venom composition between both species [13,14,27–32].

**Table 3. Interventions and outcomes of *Naja* species related injuries consulted to RECS from 2020–2023.**

| Factors | Variables | *Naja suma-trana* n (%) | *Naja kaouthia* n (%) | Undifferentiated *Naja* sp n (%) |
|---|---|---|---|---|
| Antivenom usage | No (n = 355, 69.3%) | 172 (48.4) | 95 (26.8) | 88 (24.8) |
| | Yes (n = 157, 30.7%) | 35 (22.3) | 26 (16.6) | 96 (61.1) |
| Types of antivenom | NKAV | 29 (23.6) | 24 (19.5) | 70 (56.9) |
| | NPAV | 5 (18.5) | 0 (0) | 22 (81.5) |
| | Combination (NKAV + NPAV) | 2 (33.3) | 1(16.7) | 3(50.0) |
| | Inappropriate antivenom | 1 (50.0) | 0 (0) | 1 (50.0) |
| Adverse reaction | No (n = 138, 87.9%) | 32 (23.2) | 22 (15.9) | 84 (60.9) |
| | Yes (n = 19, 12.1%) | 3 (15.8) | 4 (21.1) | 12 (63.2) |
| Type of reaction | Early anaphylactic reaction | 2 (11.8) | 4 (23.5) | 11 (64.7) |
| | Pyrogenic reaction | 1 (50.0) | 0 (0) | 1 (50.0) |
| Treatment | Hydrocortisone | 2 (15.4) | 4 (30.8) | 7 (53.8) |
| | Adrenaline | 0 | 0 | 7 |
| | Antihistamine | 2 | 2 | 2 |
| | Antipyretic | 1 (50.0) | 0 (0) | 1 (50.0) |
| Anti-tetanus toxoid | Undocumented (n = 250, 48.8%) | 129 (51.6) | 53 (21.2) | 68 (27.2) |
| | Yes (n = 262, 51.2%) | 78 (29.8) | 68 (25.9) | 116 (44.3) |
| Antibiotics | No (n = 355, 69.3%) | 157 (44.2) | 94 (26.5) | 104 (29.3) |
| | Yes (n = 157, 30.7%) | 50 (31.8) | 27 (17.2) | 80 (51.0) |
| Culture and sensitivity | No (n = 94, 59.9%) | 38 (40.4) | 17 (18.1) | 39 (41.5) |
| | Yes (n = 63, 40.1%) | 12 (19.0) | 10 (15.9) | 41 (65.1) |
| Microorganism | *Morganella morganii* | 1 (16.7) | 0 (0) | 5 (83.3) |
| | *Aeromonas* | 0 | 0 | 1 |
| | *Enterobacter* | 1 (33.3) | 0 (0) | 2 (66.7) |
| | No growth | 5 (23.8) | 3 (14.3) | 13 (61.9) |
| | Untraceable | 5 (15.6) | 5 (15.6) | 22 (68.8) |
| Surgical intervention | No (n = 398, 88.2%) | 132 (33.2) | 116 (29.1) | 150 (37.7) |
| | Yes (n = 53, 11.8%) | 14 (26.4) | 5 (9.4) | 34 (64.2) |
| Types of intervention | Wound debridement | 12 (25.0) | 5 (10.4) | 31 (64.6) |
| | Skin grafting | 2 (25.0) | 2 (25.0) | 4 (50.0) |
| | Incision and drainage | 2 | 0 | 0 |
| | Amputation | 0 | 0 | 1 |
| | Fasciotomy | 0 | 0 | 1 |
| Complications during hospital stay >24hr | No (n = 425, 91.8%) | 156 (36.7) | 109 (25.6) | 160 (37.7) |
| | Yes (n = 38, 8.2%) | 13 (34.2) | 6 (15.8) | 19 (50.0) |
| | Wound infection | 12 (35.3) | 5 (14.7) | 17 (50.0) |
| | Respiratory failure | 0 | 0 | 3 |
| | Sepsis | 1 (50.0) | 1 (50.0) | 0 (0) |
| Systemic envenomation resuscitation | No (n = 57, 76%) | 9 (15.8) | 9 (15.8) | 39 (68.4) |
| | Yes (n = 18, 24%) | 1 (5.6) | 2 (11.1) | 15 (83.3) |
| | Intubation and ventilation | 1 (4.8) | 3 (14.2) | 17 (81.0) |
| | Cardiopulmonary resuscitation | 0 | 0 | 2 |
| | Inotropes | 0 (0) | 1 (25.0) | 3 (75.0) |
| Admission to critical care | No (n = 498, 97.3%) | 205 (41.2) | 118 (23.7) | 175 (35.1) |
| | Yes (n = 14, 2.7%) | 2 (14.3) | 3 (21.4) | 9 (64.3) |

*(Continued)*

**Table 3.** (Continued)

| Factors | Variables | *Naja suma-trana* n (%) | *Naja kaouthia* n (%) | Undifferentiated *Naja* sp n (%) |
|---|---|---|---|---|
| Length of hospital stay (day) | < 1 | 38 (77.6) | 6 (12.2) | 5 (10.2) |
| | 1 | 75 (49.0) | 49 (32.0) | 29 (19.0) |
| | 2 - 3 | 37 (30.8) | 33 (27.5) | 50 (41.7) |
| | 4 - 5 | 12 (34.3) | 4 (11.4) | 19 (54.3) |
| | 6 - 10 | 9 (29.0) | 5 (16.1) | 17 (54.9) |
| | > 10 | 3 (30.0) | 0 (0) | 7 (70.0) |
| | Undocumented | 33 (28.9) | 24 (21.1) | 57 (50.0) |

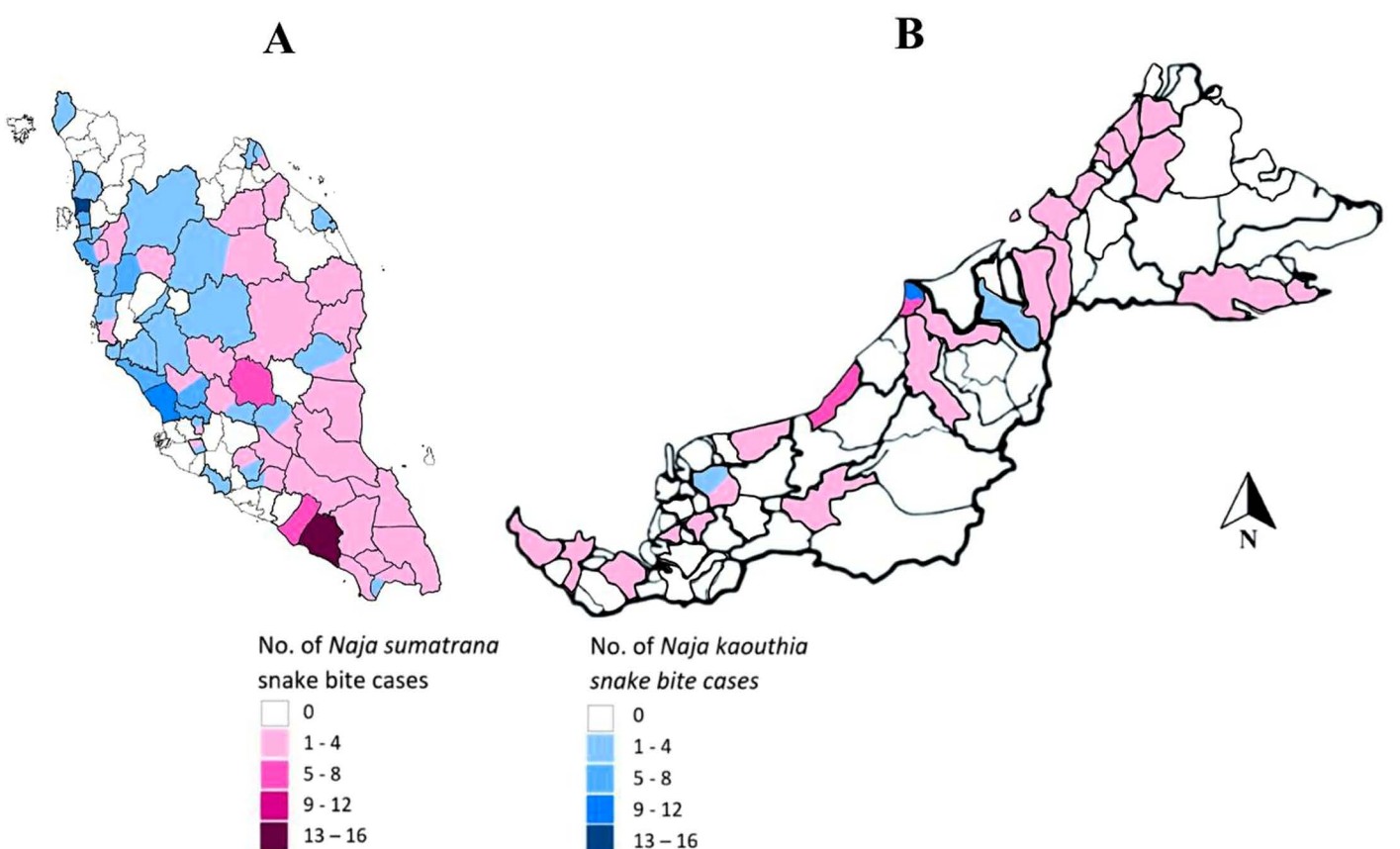

**Fig 3. Geographical distribution of confirmed *Naja spp.* related injuries patients.** Most *N. sumatrana* cases were from Johor and *N. kaouthia* cases were from Selangor. There were also *N. kaouthia* cases from Pahang, Terengganu, Johor, Negeri Sembilan and Sarawak. Base map static image and data from OpenStreetMap and OpenStreetMap Foundation. (Copyright OpenStreetMap contributors, https://www.openstreetmap.org/#map=6/4.23/108.24).

Most envenomation cases in this study received monovalent AV, rather than polyvalent or a combination of both. The appropriate antivenoms currently in use are the Thai Red Cross NKAV and Neuropolyvalent antivenom (NPAV), NKAV is primarily used to treat *N. kaouthia* envenomations and has been shown to cross-neutralize the venom of *N. sumatrana* [27–29]. These antivenoms are associated with a low incidence of adverse effects and have reduced the fatality rate

associated with envenomation by *Naja* species. The adverse effects mainly involve acute hypersensitivity reactions of varying severity, with these occurring more frequently than delayed hypersensitivity reactions. The primary toxic components in the *Naja* species venom are three-finger toxins and phospholipase A2, both of which are targeted by the Thai cobra AV. One of the three-finger toxins, alpha-neurotoxin, exhibits varying levels of abundance across different populations of *Naja* species from distinct geographical regions, meaning that the potency of the antivenom may vary depending on the incident location [27,31–33]. However, this discrepancy is not observed in this study.

There is a similar pattern of antivenom usage for *N. kaouthia* and *N. sumatrana* cases, with NKAV being the predominant choice. The less frequent use of NPAV reflects the clinical practices of determining the species prior to antivenom administration even for undifferentiated cobra bite. Patients diagnosed with an undifferentiated sp. bite were the most likely to receive antivenom, and they accounted for the majority of all antivenom doses administered. This suggests that when the specific species is unknown, clinicians may have a lower threshold for treatment. The selection of NKAV or NPAV may also be affected by the availability of the antivenom stocks in individual hospitals. This is reflected by some patients receiving both types of antivenom. This study identified two patients who received inappropriate antivenom resulting in one death. One patient was administered with Hemato Polyvalent Antivenom (HPAV) prior to RECS consultation. It was initially treated as unidentified snakebite due to unavailability of photographic evidence of the culprit snake. Following RECS consultation, the diagnosis was documented as *N. sumatrana* bite. QSMI monovalent NKAV was administered and the patient survived. In the case that resulted in death, it was diagnosed as unidentified snakebite probably from a pit viper bite and *Calloselasma rhodostoma* antivenom (CRAV) was administered without consulting RECS. RECS was consulted after death and the diagnosis was documented as undifferentiated *Naja* sp. bite envenomation.

The predominance of local over systemic manifestations is consistent with the clinical profile of Asian cobra bites described in the literature. Neurotoxicity is less prominent in Asian *Naja* sp. bites compared to African *Naja* sp, with local tissue damage being the primary concern [1]. Several studies involving Thai *Naja* sp. (*N. kaouthia, N. sumatrana* and *Naja siamensis*) have identified local envenomation as the most common clinical manifestation, particularly pain, swelling, and skin necrosis at the bite site [17,23–25,27,33]. Local tissue necrosis, often accompanied by pain and swelling, can occur even in the absence of neurological or cardiovascular dysfunctions [34–37]. These dermonecrosis may necessitate surgical interventions such as wound debridement, skin grafting, or, in extreme cases, amputation.

Systemic envenomation form *Naja* species is also associated with acute neurological dysfunction including ptosis, ophthalmoplegia, dysphagia, aphasia, hypersalivation, and, in severe cases, respiratory paralysis [10]. A study in Thailand found that most patients bitten by *Naja* sp. exhibited systemic neurological effects, with ptosis being the most common symptom, followed by muscle paralysis, bulbar palsy, dysphagia, and ophthalmoplegia [24]. Neurological symptoms were more frequently observed in patients bitten by the monocled cobra (*N. kaouthia*) compared to those bitten by undifferentiated spitting cobras (*N. sumatrana* and *N. siamensis*). *N. kaouthia* envenomation was more likely to require endotracheal intubation and ventilatory support than bites from spitting cobras. However, this study found that envenomation by *N. kaouthia* did not result in significantly more systemic effects compared to *N. sumatrana* ($p > 0.05$).

The use of antibiotics in some of the cases in this study is noteworthy. While this practice aims to prevent secondary infections, it contrasts with some recent recommendations that argued against routine antibiotic prophylaxis in snakebites, suggesting it should be reserved for cases with signs of infection [15,19,38–41]. Local effects of snakebite are often mistaken for infectious changes and can contribute to unsupported provision of antibiotics, therefore warrants careful considerations. There is no evidence supporting prophylactic antibiotics in the acute phase of clinical management unless there is evidence of secondary infection or a history of manipulation of the wound or environmental contamination [10,15,19]. The findings highlight the need for adherence to the evidence-based snakebite management guidelines in Malaysia. The generally favorable outcomes observed in this study, with low mortality are encouraging. The comparative analysis between the length of hospital stay with antivenom usage suggests that the envenomation by both species resulted in similar patterns of antivenom administration and hospitalization duration, with no significant differences detected in these

clinical outcomes. These results compare favorably with past data [22]. A study in Thailand reported a low mortality rate of 1.7%, all of which occurred in patients with *N. kaouthia* bites with varying durations of hospitalization [24]. The low mortality rate in Malaysia and Thailand reflects advancements in healthcare quality, improved support for snakebite management, and the availability and supply of appropriate antivenom [41–43]. Critically, the deaths were attributed to secondary complications that arose during hospitalization, not to untreatable, acute venom toxicity.

Complications were observed in both *N. kaouthia* and *N. sumatrana* cases and was primarily due to wound infection. Necrotizing fasciitis occurred at a higher percentage in *N. kaouthia* cases. This rate is higher than that reported in a study from Taiwan, which documented a 3.8% complication rate for *Naja atra* bites [44]. The difference could be due to variations in the snake species, environmental factors, patient characteristics, or healthcare practices. In Thailand, surgical interventions such as amputation, debridement, and skin grafting were more frequently performed in cases involving *N. kaouthia* envenomation [24]. Most patients with *N. kaouthia* bites in Thailand were asymptomatic or mildly symptomatic and were discharged after 3 to 4 days [17]. Hospitalization was significantly prolonged for patients with tissue necrosis, ranging from 20 to 29 days. These findings suggest a need to improve wound care practices. The observed discrepancies highlight the importance of further research to develop a comprehensive understanding of the differing clinical outcomes between *Naja kaouthia* and *Naja sumatrana* envenomations in Malaysia.

There appears to be a significant variability in the Bite-to-Door (BTD) and Door-to-Antivenom (DTA) administration time intervals. A DTA paradox. Bite-to-door time shows that most patients seek early medical care, often before clear indications for antivenom. However, a few patients had experienced extreme delays in reaching medical care. These were likely due to geographical barriers, limited access to emergency services, or delays in seeking help. Similarly, DTA times may highlight prolonged and inconsistent delays. However, since many patients arrived early at an emergency department, signs of systemic envenomation may not yet have been apparent. The 12-hour gap represents the necessary period of close serial observation during which the patient's progression is monitored to determine if the envenomation is severe enough to warrant AV, thereby avoiding its overuse and the risk of adverse reactions. Close serial observation of the trends in proximal progression of swelling, pain score, expansion of dermonecrosis and blood tests should become the standard practice when assessing indications for AV.

RECS management strategies emphasize accurate snake species identification, close serial observation, effective pain management, judicious use of antibiotics and appropriate and timely use of antivenom. This aligns with the current national guidelines [19]. The WHO (SEARO) recommends a similar approach, emphasizing the importance of early antivenom administration in cases of systemic envenoming or severe and progressive local effects [15]. The findings of this study highlight the need to increase public awareness on appropriate health seeking behavior, optimizing emergency response systems for faster patient transportation and streamlining hospital protocols for timely antivenom administration. These will ultimately improve patient outcomes and reduce mortality.

## Limitations

This study represents the first comprehensive, multi-year, cross-sectional analysis of *Naja* bitesconsulted to RECS and does not reflect all *Naja* sp related cases in Malaysia. There was an increase in frequency of RECS consultations to a new record high in 2023. It is uncertain whether this was due to a rise in case incidents or the popularity of RECS services among healthcare professionals. Additionally, as cases were consulted to RECS, there may be a selection bias towards more severe or complex cases, potentially overestimating the overall severity of *Naja* sp. injuries in Malaysia. The limitations can also be interpreted as a significant strength. Because the data comes from the RECS database, each case has been reviewed and verified by a clinical toxinology expert. This level of data validation is far superior to that of a standard hospital record review.

## Conclusion

*Naja* species envenomation in Malaysia is a manageable condition with low mortality but, carries a significant risk of local tissue damage. Antivenom is judiciously administered, while surgery is needed to facilitate wound healing. This study highlights the

importance of species-specific research, offering essential data for public health strategies, clinical management, and future research on snake related injuries in Malaysia. The Thai Red Cross cobra antivenom effectiveness in both species supports current antivenom strategies. Despite favorable outcomes, complication rates suggest areas for improvement in wound care and infection prevention by examining risk factors for complications and exploring interventions to reduce them in the future.

## Supporting information

**S1 Table.  Association of length of hospital stay with type of envenomation and antivenom usage of *Naja* species bites cases consulted to RECS from 2020–2023.**
(DOCX)

**S2 Table.  Comparison of antivenom usage and duration of hospitalization between the two *Naja* species cases.**
(DOCX)

**S3 Table.  Summary of mortality due to *Naja* species envenomation consulted to RECS from 2020–2023.**
(DOCX)

**S4 Table.  Comparison between the clinical pattern of injury, management, and outcome of *Naja kaouthia* and *Naja sumatrana* consulted to RECS from 2020–2023.**
(DOCX)

## Acknowledgments

Gratitude is extended to the entire team at Remote Envenomation Consultation Services (RECS) for their dedication to patient care and meticulous data collection. Acknowledgement is also given to Nur Hazwanie Abd Halim and Nurfarhana Hizan Hijas from Malaysia Biodiversity Information System (MyBIS); Nurul Saadah Ahmad and Abdul Karim Mustafa from the Department of Emergency Medicine, UKM; Khairul Hazdi Yusof from the Risk Management Unit, HCTM. The authors also acknowledge Putra Jeffri Daniel Othman, Dr. Nabil Muhammad Al Kuddoos and Muhamad Khaidir Arif Che Mat.

## Author contributions

**Conceptualization:** Ahmad Khaldun Ismail.

**Data curation:** Ahmad Khaldun Ismail, Zhi Xuan Ng, Syahirah Rezwan Eskandar, Hamelda Tanisha Ganaprakasam.

**Formal analysis:** Ahmad Khaldun Ismail, Zhi Xuan Ng, Syahirah Rezwan Eskandar, Hamelda Tanisha Ganaprakasam.

**Methodology:** Ahmad Khaldun Ismail, Zhi Xuan Ng, Syahirah Rezwan Eskandar, Hamelda Tanisha Ganaprakasam.

**Project administration:** Ahmad Khaldun Ismail.

**Supervision:** Ahmad Khaldun Ismail, Zainalabidin Mohamed Ismail.

**Validation:** Ahmad Khaldun Ismail, Zainalabidin Mohamed Ismail.

**Writing – original draft:** Ahmad Khaldun Ismail, Zhi Xuan Ng, Syahirah Rezwan Eskandar, Hamelda Tanisha Ganaprakasam.

**Writing – review & editing:** Zainalabidin Mohamed Ismail.

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
