## [Decision Letter · Decision Letter 0]

PNTD-D-25-00489

The frequency, clinical characteristics and outcomes of cobra species related injuries in Malaysia consulted to remote envenomation consultancy services from 2020-2023

Dear Dr. Ismail,

Thank you for submitting your manuscript to PLOS Neglected Tropical Diseases. After careful consideration, we feel that it has merit but does not fully meet PLOS Neglected Tropical Diseases's publication criteria as it currently stands. Therefore, we invite you to submit a revised version of the manuscript that addresses the points raised during the review process.

Please submit your revised manuscript within 60 days Jun 27 2025 11:59PM. If you will need more time than this to complete your revisions, please reply to this message or contact the journal office at plosntds@plos.org. Please include the following items when submitting your revised manuscript:

We look forward to receiving your revised manuscript.

Kind regards,

Philippe BILLIALD

Academic Editor

José María Gutiérrez

Section Editor

Shaden Kamhawi

co-Editor-in-Chief

Paul Brindley

co-Editor-in-Chief

**Additional Editor Comments:**

Thank you very much for submitting your manuscript for review by PLOS Neglected Tropical Diseases.

Your manuscript was fully evaluated by independent peer reviewers. The reviewers appreciated the attention to an important topic but identified some significant aspects of the manuscript that must be improved prior be considered for publication.

Please pay attention to their recommendations, especially those from reviewer 1

**Journal Requirements:**

1) We do not publish any copyright or trademark symbols that usually accompany proprietary names, eg ©,  ®, or TM  (e.g. next to drug or reagent names). Therefore please remove all instances of trademark/copyright symbols throughout the text, including:

- © on page: 8.

Potential Copyright Issues:

- Please confirm (a) that you are the photographer of Figure 1, or (b) provide written permission from the photographer to publish the photo(s) under our CC BY 4.0 license.

- Figure 3. Please (a) provide a direct link to the base layer of the map (i.e., the country or region border shape) and ensure this is also included in the figure legend; and (b) provide a link to the terms of use / license information for the base layer image or shapefile. We cannot publish proprietary or copyrighted maps (e.g. Google Maps, Mapquest) and the terms of use for your map base layer must be compatible with our CC BY 4.0 license.

**Reviewers' Comments:**

Reviewer's Responses to Questions

**Key Review Criteria Required for Acceptance?**

**Methods**

-Are the objectives of the study clearly articulated with a clear testable hypothesis stated?

-Is the study design appropriate to address the stated objectives?

-Is the population clearly described and appropriate for the hypothesis being tested?

-Is the sample size sufficient to ensure adequate power to address the hypothesis being tested?

-Were correct statistical analysis used to support conclusions?

-Are there concerns about ethical or regulatory requirements being met?

Reviewer #1: Objectives should be clarified.

Methods of transcribing the information from online text app in the form of consultancy and verification with case notes/case reports - need to be elaborated.

If the studies data were purely based on online text information, this should be made clear in the method description and discussed further.

Reviewer #2: Methods are well described but need some additional information.

Please include more overview of the RECS program either in the introduction or methods.

- For example: how do clinicians access the program? Do they have an app on their phone or via the computer?

- One suggestion I have is to provide more detail on inclusion and exclusion criteria of those in the study. On line 101 and 102 it mentions "alleged" cobra bite and unidentified snake-related injuries consulted to RECS were excluded. How was this determined? A simple flow map could work. For example, on line 104-106 it says the diagnosis was confirmed either by specimen brought to hospital or picture of actual specimen that caused injury or clinical presentation and syndromic approach. I think it is important to add more clarity to "clinical presentation and syndromic approach". What does that mean? Cobra is the focus of this investigation, but other venomous snake bites could be classified as cobra?

Reviewer #3: The study objectives have clearly been articulated, and the study design is appropriate to address the study objectives. Sample selection and its number is also appropriate and commendable. However, it would be beneficial if the investigators could clearly describe the difference between an alleged cobra bite and an undifferentiated cobra bite. What is the rationale behind categorizing undifferentiated cobra bites? Please explain the case definition or the mechanism used.

**Results**

-Does the analysis presented match the analysis plan?

-Are the results clearly and completely presented?

-Are the figures (Tables, Images) of sufficient quality for clarity?

Reviewer #1: See general comments below.

Reviewer #2: Results are reasonable. Tables and Figures are useful. Images of both N. sumatrana and N. Kaothia are meaningful for readers. I am very curious about "Undifferentiated Naja species bite (n=184, 36.0%)". It looks like the authors focused on the two Naja species present in Malaysia but it is unclear how the authors determined undifferentiated. Does this mean it was either sumatrana or kaothia? I would also like to see how the RECS consultants determined this and if any other "Cobra" bite occurred like those from King Cobra or possibly any outside Malaysia such as the Javan Spitting Cobra? Also were any of the envenomations due to non-native Cobra which were kept in captivity? Please comment

Reviewer #3: The results are understandable but need refinement.

Providing the nationality of snakebite victims is irrelevant, as snakebites depend on people's behavior, the snake’s behavior, and the environment, not nationality.

It's preferable to present only the most critical data instead of all findings, which may not be necessary for concluding the study. For example, occupations of victims, first aid methods, local envenoming signs, systemic envenoming signs, antibiotic usage, analgesic usage, features of acute adverse reactions to antivenom etc. (Tables 1, 2, and 3). Highlighting the most common presentations or findings is sufficient to understand the circumstances. Incorporating minor findings doesn’t influence the final picture. Or you may write them in the text without making the table too long. And too much lessor important information would deviate the attention from most important findings.

**Conclusions**

-Are the conclusions supported by the data presented?

-Are the limitations of analysis clearly described?

-Do the authors discuss how these data can be helpful to advance our understanding of the topic under study?

-Is public health relevance addressed?

Reviewer #1: See general comments below.

Reviewer #2: Discussion and conclusions are sufficient. Authors did a great job of discussing their results between the N. sumatrana and N. kaothia envenomations. Limitations were also discussed. I would suggest authors add more in regard to undifferentiated Naja species bites since other "cobra" bites may have been involved in this study. This is mentioned in lines 204-207 but likely a limitation outside the scope of RECS program.

Reviewer #3: The conclusion is well supported by the data, including a thorough description of limitations. However, the effect of a 12-hour delay in receiving antivenom may have impacted on complications and outcomes. This should be addressed more in the discussion section and conclusion. Finding a solution to reduce this delay should also be discussed in detail.

**Editorial and Data Presentation Modifications?**

Reviewer #1: (No Response)

Reviewer #2: (No Response)

Reviewer #3: Minor revisions.

**Summary and General Comments**

Reviewer #1: Authors reported a retrospective cohort study of confirmed Naja species cases consulted to Remote Envenomation Consultancy Services (RECS) from 2020 to 2023. The study aimed to identify the frequency, geographical distribution, clinical features, treatments, and outcomes of Naja species-related injuries, based on data transcribed from information in a smart text app which seemed to be the platform for the online consultancy. The finding is interesting and meaningful, especially for clinical management. However there are limitations which authors should consider to include in the manuscript, and more in-depth discussion should be made. Please see the comments per line numbers.

Line 90 - elaborate on subject matter experts (SME). What qualification does one need? Who were these experts and how the identification was verified?

Line 96-97: Authors stated in the objective to the outcomes of Naja species-related injuries in Malaysia consulted to RECS. Were the consultations provided solely by RECS online without any other consultation/opinion given by other practitioners (specifically consultants within each hospital?) which could have influenced or in one way or another contributed to the treatment and outcomes. Please provide additional information at this point as to whether the treatment outcomes were fully results from consultation to RECS only without interference.

Line 102-103: Data were extracted from "RECS consultation log and case records" - Please elaborate. What is the form of the RCES consultation log? Were these on smart text apps? How were these "record from smart texting app" transcribed into clinical data (log?), and how were these verified or cross-checked using the case records from hospitals? How many hospitals were involved and provided the primary data source which is important, how were the clinical case records reviewed and matched to the texts/data in the smart text apps?

Line 142: Query about undifferentiated Naja species... see comment below.

Line 159-160: Explain what were "undifferentiated Naja species"? Were these cases without the specimens and photos, or because of difficulty to identify? It might be better to consider stating it as "unidentified" cobra species?

Line 161: The commonest antivenom used was Thai Red Cross Naja kaouthia monospecific antivenom... (78.3%). Include more information regarding the remaining 22.7%? What antivenoms were given? How many did not receive antivenom?

Line 168-172: Provide details of complications: what complications specifically in the 38 (8.2%) patients? Of these, 34 were due to wound infection, what about the other 4? Please also provide details of time and severity of the complications/wound infection or local effect?

Most frequent emergency resuscitation was for airway intubation and ventilation (n=21): Provide details, the time (onset), and the causative species and distribution?

Line 180: Include comparative analysis for different antivenom usage? The majority were treated with the Thai Naja kaouthia monospecific antivenom and how about the others?

Line 190: Four deaths were due to unidentified Naja species bite. What attempts had been to identify the species, did the patients received any treatment including ventilation and antivenom (what antivenom)? Any difference between "unidentified" and "undifferentiated" species?

Line 196-197: It is crucial to state clearly whether this is a hospital based study or purely based on smart text apps-consultations. If this was the case (data fully derived from smart text apps on mobile phones), even though there were some statistical analysis of the "consultations" transcribed from the online consultation apps, authors should discuss the limitation especially for not able to capture cases not consulted to the RECS apps. Although the number of case was large, it is debatable how representative the finding is for the Naja bite in the whole Malaysia. It would be good to be more conservative in interpreting the results, which were unconventional to hospital-based epidemiological study and retrospective survey.

Line 359-361: The difference in local envenoming severity....stress the need for species-specific management protocols. -This is true but in the result and explanation are not in-depth. Please provide more detailed comparison between the two, and if possible, representative images of the findings.

Reviewer #2: Thank you for presenting your findings from the RECS program for two highly venomous cobra species found in Malaysia. The manuscript is written well, and data is important for ongoing surveillance practices of snake bite in this region of the world. Authors are experts in their field and have provided a well written manuscript.

In addition to the comments provided above I think it will be important to consider not generalizing the term "Cobra" in the title as there are additional species found in Malayasia and Borneo itself where cobra is used as a common name. Authors do not comment on "King Cobra", Ophiophagus hannah bites. Did any occur in this cohort? The Javan Spitting Cobra (Naja sputatrix) found in Borneo, albeit found in the southern regions of Indonesia on islands of Java and Lesser Sunda, is another species that may be involved in this cohort. Do we know if any included in this cohort were bitten by these species? Also, do we know if anyone had traveled from a region within Indonesia and back to Malaysia for care for cobra snake bite? My major recommendation is to ensure the RECS program is described well in the introduction and methods section. I would also recommend to the authors that they define "undifferentiated Naja species" better and whether any of these snakes could be non-native Naja from other regions of Asia or even King Cobra bites.

Reviewer #3: While many similar studies exist globally, studying geographically unique epidemiological data, clinical presentations, complications, and management of snakebites is crucial for improving patient outcomes. In this study, authors should focus more on presenting only the relevant data rather than including excessive information that makes the manuscript overly long.

PLOS authors have the option to publish the peer review history of their article (what does this mean? ). If published, this will include your full peer review and any attached files.

**Do you want your identity to be public for this peer review?** For information about this choice, including consent withdrawal, please see our Privacy Policy .

Reviewer #1: No

Reviewer #2: **Yes: ** Norman L. Beatty, University of Florida College of Medicine, Gainesville, FL USA

Reviewer #3: No

**Figure resubmission:**
---

## [Editor Report · Decision Letter 1]

Dear Dr Ismail,

We are pleased to inform you that your manuscript 'The frequency, clinical characteristics and outcomes of Naja species related injuries in Malaysia consulted to remote envenomation consultancy services from 2020-2023' has been provisionally accepted for publication in PLOS Neglected Tropical Diseases.

Best regards,

Philippe BILLIALD

Academic Editor

José María Gutiérrez

Section Editor

Shaden Kamhawi

co-Editor-in-Chief

Paul Brindley

co-Editor-in-Chief

---

## [Editor Report · Acceptance letter]

Dear Dr Ismail,

We are delighted to inform you that your manuscript, "The frequency, clinical characteristics and outcomes of Naja species related injuries in Malaysia consulted to remote envenomation consultancy services from 2020-2023," has been formally accepted for publication in PLOS Neglected Tropical Diseases.

Best regards,

Shaden Kamhawi

co-Editor-in-Chief

Paul Brindley

co-Editor-in-Chief
